# Dark Chocolate Mitigates Premenstrual Performance Impairments and Muscle Soreness in Female CrossFit^®^ Athletes: Evidence from a Menstrual-Phase-Specific Trial

**DOI:** 10.3390/nu17081374

**Published:** 2025-04-18

**Authors:** Kousar Safari, Mohammad Hemmatinafar, Katsuhiko Suzuki, Maryam Koushkie Jahromi, Babak Imanian

**Affiliations:** 1Department of Sport Sciences, Faculty of Education and Psychology, Shiraz University, Shiraz 84334-71946, Iran; 2Faculty of Sport Sciences, Waseda University, 2-579-15 Mikajima, Tokorozawa 359-1192, Japan

**Keywords:** dark chocolate, cognitive performance, menstrual cycle, premenstrual syndrome, CrossFit^®^, DOMS, Stroop test, high-intensity training, female athletes

## Abstract

**Background:** Hormonal fluctuations across the menstrual cycle can significantly impair physical performance, neuromuscular function, and cognitive processing in female athletes, particularly during the premenstrual phase. Emerging evidence suggests that dark chocolate (DC), rich in flavonoids, polyphenols, magnesium, and theobromine, may exert anti-inflammatory, analgesic, and neuroprotective effects. This study investigated the acute effects of 85% DC supplementation on cognitive and physical performance, as well as delayed-onset muscle soreness (DOMS), in female CrossFit^®^ athletes across four distinct hormonal phases. **Methods:** In this randomized, double-blind, placebo-controlled, crossover study, fifteen trained eumenorrheic female CrossFit^®^ athletes completed three intervention conditions: dark chocolate (DC), placebo (PLA), and control (CON). Participants were evaluated during four distinct menstrual phases—menstrual, follicular, luteal, and premenstrual syndrome (PMS)—over three consecutive menstrual cycles. In each phase, participants consumed 30 g/day of either DC or PLA for three days, followed by performance testing on day four. Functional and cognitive performance were assessed via the CINDY WOD, handgrip strength (HGS), and Stroop tests (reaction time and correct answer percentage, CAP). DOMS was measured using a 100 mm visual analog scale at baseline and at 0, 12, 24, 48, and 72 h post-exercise. **Results:** DC supplementation significantly improved functional performance (CINDY WOD) across all menstrual phases, with the greatest enhancement during PMS (*p* < 0.01). Reaction time significantly improved during PMS (*p* = 0.010 vs. control; *p* = 0.002 vs. placebo). Additionally, DOMS was notably reduced in the luteal phase at 12 h, 24 h, and 72 h post-exercise in the DC condition compared to the control and placebo (*p* < 0.05). No significant differences were observed in HGS across conditions or phases (*p* > 0.05). **Conclusions:** Short-term DC supplementation may selectively enhance high-intensity functional performance and cognitive accuracy in trained female athletes, particularly during hormonally sensitive phases such as PMS. Its anti-inflammatory and neuromodulatory properties make DC a promising, non-pharmacological strategy to support female-centric recovery and performance in CrossFit^®^ and similar sports. Future research should explore chronic intake, mechanistic biomarkers, and individual variability across menstrual cycles.

## 1. Introduction

CrossFit^®^ is recognized as one of the fastest-growing forms of high-intensity functional training (HIFT) [1]. This strength and conditioning program aims to optimize physical competence across ten fitness domains: cardiovascular/respiratory endurance, stamina, strength, flexibility, power, speed, coordination, agility, balance, and accuracy [2]. Training typically involves the “workout of the day” (WOD)—a combination of functional, high-intensity exercises executed repetitively and rapidly with minimal rest between sets [3]. While this structure provides comprehensive performance benefits, it can also lead to exercise-induced muscle damage (EIMD)—a phenomenon linked to excessive mechanical and metabolic stress, which often exceeds the adaptive capacity of skeletal muscle [4]. EIMD is characterized by structural disruption, mitochondrial dysfunction, and an inflammatory cascade resulting in muscle soreness, swelling, and decreased contractile strength [5]. It is associated with increased levels of muscle damage biomarkers such as creatine kinase (CK), lactate dehydrogenase (LDH), and myoglobin (Mb) [6], alongside systemic elevations in C-reactive protein (CRP), interleukins (IL-1, IL-6), and tumor necrosis factor-alpha (TNF-α) [7]. These responses compromise recovery and athletic performance—especially in female athletes, where hormonal fluctuations during the menstrual cycle may exacerbate inflammation, pain sensitivity, and neuromuscular fatigue [8].

Hormonal fluctuations throughout the menstrual cycle influence key physiological systems. While resistance exercise stimulates acute anabolic hormonal responses—including increased testosterone and growth hormone levels following exercise—fluctuations of estrogen and progesterone across the cycle phases uniquely impact female physiology [9,10]. Elevated estrogen in the follicular phase supports improved substrate metabolism and thermoregulation. In contrast, increased progesterone during the luteal and premenstrual phases is associated with reduced neuromuscular coordination, higher body temperature, substantial fatigue, and pain sensitivity [11]. These cyclical effects explain why women may experience variable performance outcomes depending on their hormonal state. Moreover, premenstrual syndrome (PMS)—manifesting as mood swings, irritability, reduced cognitive function, and physical discomfort—can significantly hinder training output [12]. HIFT, like CrossFit^®^, induces complex aerobic and anaerobic adaptations, including enhanced mitochondrial capacity, neuromuscular efficiency, and lactate tolerance [13]. These adaptations improve overall fitness and increase oxidative stress and inflammation, particularly in female athletes experiencing menstrual-cycle-related hormonal fluctuations [14]. Nutritional strategies that mitigate these responses are critical for optimizing recovery and performance. Recent findings by Moscatelli et al. (2023) emphasize the metabolic and neuromuscular demands of CrossFit^®^ training [13], supporting the relevance of targeted interventions—such as dark chocolate (DC) supplementation—to supporting physiological resilience and performance during hormonally challenging phases.

Given the multifactorial effects of the menstrual cycle on athletic readiness, nutrition-based strategies have been proposed to mitigate EIMD and improve recovery in female athletes [15,16,17,18]. In this context, DC—rich in cocoa-derived flavonoids, polyphenols, and micronutrients—has emerged as a promising functional food with ergogenic potential [19]. Cocoa flavonoids such as epicatechin and catechin possess well-documented antioxidant, anti-inflammatory, vasodilatory, and neuroprotective properties, which may aid in reducing oxidative damage, improving blood flow, and alleviating muscle soreness [20]. Emerging evidence indicates that DC, particularly its cocoa flavanol content, can positively influence glucose–insulin dynamics and enhance cerebral blood flow during physical activity, potentially improving metabolic efficiency and supporting cognitive performance [21]. Its polyphenols, particularly cocoa flavanols, have been shown to enhance mitochondrial function, promote endothelial nitric oxide production, and increase capillary density—physiological adaptations that support improved endurance performance and aid in exercise recovery [22,23]. DC further contains magnesium, iron, theobromine, and trace amounts of omega-3 and -6 fatty acids, which may be beneficial in relieving menstrual pain and premenstrual symptoms through anti-spasmodic and mood-stabilizing effects [24,25].

Despite these theoretical and preliminary benefits, the effects of DC supplementation on athletic and cognitive performance across menstrual cycle phases remain underexplored. Previous studies have investigated DC’s potential to alleviate menstrual pain and enhance physical or cognitive performance in trained women [24,26]; however, the underlying mechanisms and practical applications in sport-specific settings require further investigation. Additionally, the interaction between DC, hormonal status, and neuromuscular function during high-intensity functional training remains poorly understood. Notably, while research has examined DC’s antioxidant and anti-inflammatory effects [19,21,27], very few studies have examined its impact on physical and cognitive performance across the hormonal spectrum of the menstrual cycle in athletic women, particularly during the menstrual cycle and PMS, which are known to impair training efficiency [8,12,28]. Understanding this relationship may provide novel strategies to support recovery, reduce muscle soreness, and maintain performance in female athletes participating in high-stress sports, such as CrossFit^®^.

Despite these emerging insights, current research on the ergogenic effects of DC across the menstrual cycle remains limited, particularly in high-intensity training modalities. Notably, the premenstrual phase—characterized by increased inflammation, reduced neuromuscular efficiency, cognitive disruptions, and heightened pain sensitivity—poses a substantial barrier to optimal training in female athletes. While a few studies have explored the role of DC in modulating pain and enhancing mental performance, very few have examined its comprehensive physiological and cognitive impacts during hormonally sensitive phases such as PMS. This represents a critical knowledge gap, as PMS symptoms have been associated with compromised recovery, increased DOMS, and reduced training efficacy. Furthermore, we could not find any studies that have assessed whether acute DC supplementation can alleviate these limitations during sport-specific functional testing in trained female populations.

Therefore, the present study investigates the acute effects of 85% DC supplementation on muscle soreness, perceived pain, cognitive performance, handgrip strength, and sports performance across distinct menstrual phases, particularly emphasizing the premenstrual phase. By employing a randomized, placebo-controlled, double-blind crossover design among experienced female CrossFit^®^ athletes, this study aims to clarify whether DC can serve as a non-pharmacological strategy to enhance athletic performance and psychological readiness during hormonally challenging phases. The findings may yield practical applications for female-centered sports nutrition and contribute to personalized recovery strategies in high-stress, performance-driven environments.

## 2. Methods

### 2.1. Participants

Fifteen trained female CrossFit^®^ athletes (mean competition experience: 3.0 ± 0.8 years) voluntarily participated in this study. Table 1 contains the demographic information of the participants. Participants were initially screened six months prior to data collection using the Premenstrual Symptoms Screening Tool (PSST), a validated instrument for assessing both menstrual cycle regularity and the presence and severity of premenstrual syndrome (PMS) symptoms in women [29]. A total of 50 CrossFit^®^ athletes were considered, but 35 were excluded due to irregular menstrual cycles. Moreover, based on the daily monitoring and tracking of basal body temperature and symptom logs, only participants with confirmed regular menstrual cycles and a minimum cycle length of 28 days were included. For this study, the menstrual cycle was divided into four distinct physiological phases: (1) the menstrual phase (M), defined as the duration of active uterine bleeding; (2) the follicular phase (F), extending from the end of menstruation to the onset of the luteal phase; (3) the luteal phase (L), occurring post-ovulation and ending five days before menstruation; and (4) the premenstrual syndrome phase (PMS), defined as the final 3–5 days before menstruation, characterized by both emotional and somatic symptoms [30,31]. Participants refrained from consuming chocolate products, caffeine, alcohol, tobacco, or medications that might disrupt hormonal balance during data collection. All athletes followed identical training regimens under professional supervision and were instructed to avoid strenuous activity for 48 h before and after test days. The study protocol was thoroughly explained before enrollment, and written informed consent was obtained. This study was approved by The Research Ethics Committees of the Faculty of Psychology and Educational Sciences, Shiraz University (Approval Code: SEP.14023.48.5571, 2023), and conducted in accordance with the principles outlined in the Declaration of Helsinki.

### 2.2. Sample Size Calculation and Study Design

The sample size was calculated using the G*Power analysis software (version 3.1.9.7) [32] based on a 5% Type I error rate (α), 0.80 statistical power (1 − β), and a 0.90 correlation. Drawing on data from a previous DC study [27], with an effect size (Δ of response) of 0.5 for the total repetition variable, the calculated sample size was 12 participants. To account for potential dropout and enhance the robustness of our findings, we increased the sample size to 15 participants.

This study employed a randomized, double-blind, placebo-controlled, crossover design to evaluate the short-term effects of DC supplementation on exercise-induced muscle soreness, physical performance, and cognitive function across four distinct phases of the menstrual cycle in trained female CrossFit^®^ athletes (Figure 1). The study protocol was implemented in accordance with established guidelines for sports nutrition research design and methodological rigor [33,34,35]. Prior to the intervention, participants attended a standardized familiarization session to practice all cognitive and physical performance assessments, including the Stroop test [36], handgrip strength (HGS) test [37], and Cindy’s workout of the day (CINDY WOD) [38]. Fifteen athletes were randomly assigned to one of three intervention conditions: dark chocolate (DC, *n* = 5), placebo (PLA, *n* = 5), or control (CON, *n* = 5). Three intervention periods were conducted, each lasting for one complete menstrual cycle (28 days), during which participants were tested in four hormonally distinct phases: M, F, L, and PMS. Phase classification was determined using the PSST and confirmed by daily basal body temperature tracking [29]. During each phase, participants consumed either 30 g/day of 85% DC or a visually and nutritionally matched placebo (low-cocoa chocolate) for three consecutive days, and the fourth day was designated as the test day, during which they performed the Stroop, HGS, and CINDY WOD tests in that order at 10 min intervals. DOMS was assessed using a 100 mm visual analog scale (VAS) at baseline, 0, and at 12, 24, 48, and 72 h following CINDY WOD [39,40]. A two-month washout period was imposed between intervention phases to prevent carryover effects. All testing was conducted between 9:00 AM and 1:00 PM to control for diurnal variation, and participants consumed a standardized breakfast (~250 kcal: 45 g carbohydrates, 9 g protein, and 5 g fat) 90 min before testing [41]. Water intake during testing was restricted and recorded. Participants were instructed to maintain a consistent diet, avoid consuming additional food within one hour before each test, and refrain from strenuous activity for 48 h preceding each test day.

### 2.3. Supplementation Procedures

In accordance with this study’s randomized, double-blind, placebo-controlled crossover design, participants were assigned to one of three experimental conditions: DC, PLA, or CON. Each participant consumed the assigned supplement—either 30 g/day of 85% DC (Lindt Excellence, Kilchberg, Switzerland) [42] or a visually and calorically matched placebo chocolate (PLA)—for three consecutive days at each menstrual cycle phase, followed by a test day on the fourth day (Figure 2). Participants in the CON condition did not receive any supplement and were evaluated under natural, unsupplemented conditions. The DC used in the DC condition contained 588 kcal per 100 g and consisted of 70.8% fat, 21.6% protein, 10.98% carbohydrate, and naturally occurring flavonoids and polyphenolic compounds [43]. To maintain the integrity of the double-blind design, the placebo chocolate was explicitly formulated to closely match the DC in appearance, texture, flavor, caloric content, and macronutrient composition but contained less than 10% cocoa solids. Both the DC and PLA supplements were provided in identical, unlabeled packaging and carefully matched in color and taste to ensure adequate blinding of participants and researchers alike [33].

All participants were instructed to refrain from consuming additional chocolate products, caffeine, or polyphenol-containing supplements throughout the intervention. Each participant completed all three intervention arms (DC, PLA, and CON) across different menstrual phases using a counterbalanced crossover protocol, with a two-month washout period between conditions to eliminate potential carryover effects. The supplementation procedure was designed to ensure physiological stability, control for hormonal fluctuations, and optimize methodological consistency across all test weeks.

### 2.4. CINDY WOD

The CrossFit^®^ benchmark workout known as “Cindy” served as the functional performance assessment in this study. Cindy is a time-based workout that requires participants to complete as many rounds as possible (AMRAP) in 20 min of the following sequence: 5 pull-ups, 10 push-ups, and 15 air squats. Participants were instructed to perform the exercises continuously and in the prescribed order, completing all repetitions of a given movement before advancing to the next. Performance was quantified by the total number of full rounds completed plus additional repetitions from any final incomplete round, if applicable. To ensure test validity and consistency among participants, trained observers strictly enforced movement standards. Pull-ups were considered valid when the participant started with arms fully extended and completed the repetition with the chin clearly passing above the bar before returning to full arm extension. Participants were permitted to use strict, kipping, or butterfly variations in accordance with standard CrossFit^®^ regulations. Participants maintained a plank position for push-ups with hands placed directly beneath the shoulders, lowering the chest to touch the ground and then returning to full arm extension. Air squats were deemed valid when the hip crease dropped below the plane of the knee, and the participant returned to a fully upright position. Any failure to meet these criteria resulted in the repetition being disqualified and needing to be repeated to count toward the final score. This protocol has been validated in previous CrossFit^®^-based performance studies as a reliable measure of muscular endurance, metabolic conditioning, and functional capacity in trained individuals [44].

### 2.5. Examination of Delayed-Onset Muscle Soreness by the VAS

The VAS measured the amounts of DOMS. On this scale, a horizontal line of 100 mm is drawn, at the beginning of which the phrase is painless and at the end of which the word is severe pain [39]. The VAS is a number that allows a person to express the severity of their pain and is used in experimental and clinical studies. The subjects determined their perception of the severity of DOMS before the start of the test (DOMS baseline), immediately after (DOMS 0 h), and 12 (DOMS 12 h), 24 (DOMS 24 h), 48 (DOMS 48 h), and 72 (DOMS 72 h) hours after CINDY WOD [45]. By measuring the distance of points marked from the line’s origin with a ruler, the pain score of each person was recorded in centimeters [40]. These time points were chosen based on established research indicating that muscle soreness typically begins within 12 h of high-intensity eccentric exercise, peaks around 24–72 h, and gradually resolves after that [46]. This framework allowed for a comprehensive assessment of the progression and peak of DOMS following the intervention.

### 2.6. Handgrip Strength Test (HGS)

HGS, a validated indicator of upper body isometric force and neuromuscular function, was assessed using a calibrated hydraulic hand dynamometer (SH5001 Hand Dynamometer, 200 lbf/90 kg capacity, Lügde, Germany). Participants were seated with the elbow flexed at 90°, and measurements were taken with the non-dominant hand unless physical limitation required using the dominant hand. After instructions were provided, each participant performed three maximal isometric contractions, with a one-minute rest between trials, and the highest value was recorded. The testing order was randomized, and participants were advised to stop immediately if pain occurred. This procedure followed validated guidelines for isometric strength assessment in exercise science [47].

### 2.7. Stroop Test (ST)

The Stroop test was conducted in a controlled sports science laboratory under standardized environmental conditions, with ambient temperature ranging from 22 to 24 °C and relative humidity between 4 and 15% across sessions. During the familiarization session, all participants were introduced to the test protocol, which included reaction time (RT) and Stroop task procedures, to minimize learning effects. Participants performed three RT trials, each separated by a 30 s rest interval, using a digital computer-based version of the Stroop task. Before the main task, six practice trials were administered, followed by 40 test trials. The accuracy rate ranged from 80% to 98%. For the RT task, participants were instructed to maintain their index finger in contact with the spacebar and to respond as quickly as possible when a green spherical stimulus (10 cm in diameter) appeared on the screen. The screen-to-eye distance was maintained at approximately 55 cm, and reaction times were recorded in milliseconds using embedded software and exported to an Excel data sheet in real-time. Reaction time in this context reflects an individual’s attention and processing speed, representing the latency between the presentation of a visual stimulus and the participant’s motor response. This digital version of the Stroop test is widely used in sport and exercise science to assess cognitive flexibility, selective attention, and executive control under exercise or supplementation conditions [48,49]. This test was performed once before CINDY WOD and RT, and the correct answer percentage (CAP) was recorded as part of the Stroop test’s results.

### 2.8. Data Analysis

Data were analyzed using IBM SPSS Statistics version 26.0 (IBM Corp., Armonk, NY, USA). The Shapiro–Wilk test was employed to assess the normality of data distribution. A one-way repeated measures ANOVA was used to analyze differences in functional performance across conditions. A two-way repeated measures ANOVA (condition × time) was applied to analyze changes in DOMS scores (VAS) across time points and conditions. Where significant main or interaction effects were observed, Bonferroni-adjusted post hoc comparisons were conducted to determine pairwise differences. The partial eta squared (pEta2) was calculated as an effect size measure for interaction and main effects. According to Cohen, pEta2 ≥ 0.01 indicates small effects, pEta2 ≥ 0.059 indicates medium effects, and pEta2 ≥ 0.138 indicates large effects [50]. Statistical significance was set at *p* ≤ 0.05, and all results are reported as mean ± standard deviation (SD). Figures were generated using GraphPad Prism version 9.0.0 (GraphPad Software, San Diego, CA, USA).

## 3. Results

The descriptive characteristics of the functional tests are presented in Table 2, while the means and standard deviations of the DOMS assessments are provided in Table 3.

### 3.1. Menstrual Phase

Statistical analysis revealed that the main effect of the intervention on CINDY WOD was significant (F_2,00_ = 4.71, *p* = 0.017, pEta^2^ = 0.252). Post hoc Bonferroni tests indicated that participants in the DC condition exhibited significantly higher performance in the CINDY WOD test compared to the CON (*p* = 0.017). However, no significant differences were observed between the DC and PLA (*p* = 0.513) and PLA and CON (*p* = 0.433) (Table 4) (Figure 3, menstrual phase).

The main effect of the intervention was not significant for HGS (F_2,00_ = 0.81, *p* = 0.452, pEta^2^ = 0.055), RT (F_2,00_ = 1.98, *p* = 0.156, pEta^2^ = 0.124), or CAP (F_2,00_ = 2.51, *p* = 0.099, pEta^2^ = 0.152) (Table 4) (Figure 3, menstrual phase).

For the DOMS, the results showed that the main effect was considerable (F_2,78_ = 22.51, *p* = 0.001, pEta^2^ = 0.617), and the results of the Bonferroni test indicated that in the CON condition, DOMS 12 h and 24 h was significantly higher compared to the baseline (*p* = 0.021, *p* = 0.024), DOMS 48 h (*p* = 0.031, *p* = 0.012), and DOMS 72 h (*p* = 0.008, *p* = 0.004). In the PLA condition, DOMS 12 h was substantially higher compared to the baseline (*p* = 0.002), DOMS 48 h (*p* = 0.002), and DOMS 72 h (*p* = 0.049). Also, DOMS 24 h (*p* = 0.024) and DOMS 48 h (*p* = 0.049) were significantly higher compared to the baseline. Nevertheless, in the DC condition, there were no considerable differences between the times of the DOMS (*p* > 0.05). Moreover, the results of the post hoc test showed that DOMS 12 h in the DC condition was significantly lower compared to the CON (*p* = 0.047). However, no significant difference was observed between the three conditions of the study in the baseline and DOMS 0 h, 24 h, 48 h, and 72 h (*p* > 0.05). All of these results are shown in the Figure 4 menstrual phase graph.

### 3.2. Follicular Phase

Statistical analysis revealed that the main effect of the intervention on CINDY WOD was significant (F_2,00_ = 7.67, *p* = 0.002, pEta^2^ = 0.354). Post hoc Bonferroni tests indicated that participants in the DC condition exhibited significantly higher performance in the CINDY WOD test compared to the CON (*p* = 0.008). However, no significant differences were observed between the DC and PLA (*p* = 0.077) and PLA and CON (*p* = 0.307) (Table 4) (Figure 3, follicular phase).

The main effect of the intervention was not significant for HGS (F_1,33_ = 3.52, *p* = 0.066, pEta^2^ = 0.201), RT (F_1,28_ = 0.04, *p* = 0.890, pEta^2^ = 0.003), and CAP (F_2,00_ = 1.35, *p* = 0.273, pEta^2^ = 0.088) (Table 4) (Figure 3, follicular phase).

For the DOMS, the results revealed that the main effect was considerable (F_3,86_ = 22.67, *p* = 0.001, pEta^2^ = 0.618), and the results of the Bonferroni test indicated that in the CON condition, DOMS 12 h and 24 h was significantly higher compared to the baseline (*p* = 0.043, *p* = 0.005) and DOMS 72 h (*p* = 0.020, *p* = 0.002). Additionally, DOMS 24 h was considerably higher compared to DOMS 48 h (*p* = 0.021). In the PLA condition, DOMS 12 h was substantially higher compared to the baseline (*p* = 0.022), DOMS 48 h (*p* = 0.025), and DOMS 72 h (*p* = 0.026). Also, DOMS 24 h was significantly higher compared to DOMS 48 h (*p* = 0.024). Nevertheless, in the DC condition, there were no considerable differences between the times of the DOMS (*p* > 0.05). However, the results of the post hoc test showed that there were no significant differences between the three conditions of the study in each time of the DOMS (*p* > 0.05). All of these results are shown in Figure 4 in the follicular phase graph.

### 3.3. Luteal Phase

Statistical analysis revealed that the main effect of the intervention on CINDY WOD was significant (F_2,00_ = 10.65, *p* = 0.001, pEta^2^ = 0.432). Post hoc Bonferroni tests indicated that participants in the DC condition exhibited significantly higher performance in the CINDY WOD test compared to the CON (*p* = 0.001). However, no significant differences were observed between the DC and PLA (*p* = 0.213) and PLA and CON (*p* = 0.096) (Table 4) (Figure 3, luteal phase).

The analysis further demonstrated that the main effect of the intervention on CAP was significant (F_2,00_ = 3.73, *p* = 0.036, pEta^2^ = 0.211). Post hoc tests revealed a substantial increase in CAP in the DC condition compared to the CON (*p* = 0.040). Nevertheless, no significant difference was noted between DC and PLA (*p* = 0.108) and PLA and CON (*p* = 1.000) (Table 4) (Figure 3-Luteal phase).

The main effect of the intervention was not significant for HGS (F_2,00_ = 1.82, *p* = 0.180, pEta^2^ = 0.115) and RT (F_2,00_ = 1.22, *p* = 0.310, pEta^2^ = 0.080) (Table 4) (Figure 3, luteal phase).

For the DOMS, the results showed that the main effect was considerable (F_2,19_ = 23.32, *p* = 0.001, pEta^2^ = 0.625), and the results of the Bonferroni test indicated that in the CON condition, DOMS 12 h was significantly higher compared to the baseline (*p* = 0.001), DOMS 0 h (*p* = 0.001), DOMS 48 h (*p* = 0.011), and DOMS 72 h (*p*= 0.003). Additionally, DOMS 24 h was considerably higher compared to the baseline (*p* = 0.001) and DOMS 72 h (*p* = 0.008). Moreover, DOMS 48 h was significantly higher compared to the baseline (*p* = 0.020). In the PLA condition, DOMS 12 h and 24 h were substantially higher compared to the baseline (*p* = 0.032, *p* = 0.013) and DOMS 72 h (*p* = 0.003, *p* = 0.008). In the DC condition, DOMS 12 h was considerably higher compared to the baseline (*p* = 0.026), DOMS 48 h (*p* = 0.043), and DOMS 72 h (*p* = 0.008). Additionally, DOMS 24 h was significantly higher compared to the baseline (*p* = 0.038) and DOMS 72 h (*p* = 0.006). Moreover, the results of the post hoc test showed that DOMS 12 h (*p* = 0.042) and DOMS 24 h (*p* = 0.029) in the DC condition were substantially lower compared to the CON. Also, DOMS 72 h considerably decreased in the DC condition compared to the PLA (*p* = 0.026) and CON (*p* = 0.045). All of these results are shown in Figure 4 in the luteal phase graph.

### 3.4. Premenstrual Phase

Statistical analysis revealed that the main effect of the intervention on the CINDY WOD was significant (F_2,00_ = 10.27, *p* = 0.001, pEta^2^ = 0.423). The Bonferroni tests indicated that participants in the DC condition exhibited significantly higher performance in the CINDY WOD test compared to the CON (*p* = 0.002). However, no significant differences were observed between the DC and PLA (*p* = 0.157) and PLA and CON (*p* = 0.087) (Table 4) (Figure 3, premenstrual phase).

The analysis further demonstrated that the main effect of the intervention on reaction time was significant (F_2,00_ = 9.50, *p* = 0.001, pEta^2^ = 0.404). Post hoc tests revealed a substantial decrease in reaction time in the DC condition compared to the PLA (*p* = 0.002) and CON (*p* = 0.010). Nevertheless, no significant difference was noted between PLA and CON (*p* = 1.000) (Table 4) (Figure 3, premenstrual phase).

The main effect of the intervention was not significant for HGS (F_2,00_ = 0.44, *p* = 0.648, pEta^2^ = 0.030) and CAP (F_2,00_ = 0.18, *p* = 0.832, pEta^2^ = 0.013) (Table 4) (Figure 3, premenstrual phase).

For the DOMS, the results showed that the main effect was considerable (F_2,43_ = 23.23, *p* = 0.001, pEta^2^ = 0.624), and the results of the Bonferroni test indicated that in the CON condition, DOMS 12 h (*p* = 0.018) and 24 h (*p* = 0.028) were significantly higher compared to the baseline. Additionally, DOMS 24 h was substantially higher compared to DOMS 72 h (*p* = 0.035). In the PLA condition, DOMS 0 h was significantly higher compared to the baseline (*p* = 0.040). In the DC condition, DOMS 0 h was substantially higher compared to the baseline (*p* = 0.028), DOMS 48 h (*p* = 0.020), and DOMS 72 h (*p* = 0.008). In addition, DOMS 12 h was considerably higher compared to the baseline (*p* = 0.014), DOMS 48 h (*p* = 0.005), and DOMS 72 h (*p* = 0.004). Moreover, DOMS 24 h was significantly higher compared to DOMS 72 h (*p* = 0.033). However, the results of the post hoc test showed that there were no significant differences between the three conditions of the study at each time of the DOMS (*p* > 0.05). All of these results are shown in Figure 4 in the premenstrual phase graph.

## 4. Discussion

### 4.1. Overview of Main Findings

In this randomized controlled trial, short-term DC supplementation resulted in significant improvements in specific physical and cognitive performance outcomes among female CrossFit^®^ athletes. In particular, DC intake enhanced high-intensity functional exercise performance, as evidenced by higher scores in the CINDY WOD across all phases of the menstrual cycle compared to placebo and control conditions. DC’s benefits were especially pronounced in the late luteal (premenstrual) phase, where it significantly improved cognitive function—participants demonstrated faster reaction times and greater accuracy on the Stroop test—relative to their performance with placebo or no supplementation. In contrast, DC had minimal impact on maximal isometric strength (HGS) in any phase, and its effect on exercise-induced muscle soreness was limited. Specifically, DOMS showed a modest reduction with DC only during the luteal phase, with no significant DC-related relief observed in the menstrual, follicular, or premenstrual syndrome phases. Together, these findings suggest that while acute DC consumption can selectively enhance endurance-type workout performance and certain cognitive aspects in female athletes, these ergogenic effects are context-dependent, manifesting variably across the menstrual cycle. The late luteal phase, often associated with PMS symptoms, emerged as a window in which DC conferred the greatest overall benefit, particularly for cognitive performance. Meanwhile, outcomes such as muscular strength remained unchanged by DC, underscoring that its ergogenic influence may be more relevant for metabolic and neuromuscular endurance aspects of performance rather than maximal force production.

### 4.2. Phase-Specific Performance Across Menstrual Cycle Phases

Menstrual phase: During the menstrual phase, characterized by low estrogen and progesterone levels, DC-supplemented athletes demonstrated significantly improved high-intensity exercise performance compared to the CON condition. This suggests that DC provided an ergogenic boost even when baseline performance might be hindered by menstrual symptoms such as pain or fatigue. Menstruation can entail cramping and lethargy that impair exercise capacity, and nutritional interventions are often sought to counter these effects. Verma and Kadam [24] reported that daily DC intake significantly reduced menstrual pain in young women, likely by modulating central neurotransmitters, thereby improving comfort during this phase. The enhanced CINDY WOD performance we observed with DC may stem from such effects—reduced pain perception and improved mood—as cocoa is known to promote the activity of endorphins and serotonin in the brain. Consistent with this, one study noted that DC consumption can serve as a non-pharmacological alternative for relieving dysmenorrhea (menstrual pain) in women [25]. On the other hand, we found no significant changes in handgrip strength or reaction time due to DC during menses, indicating that maximal strength and simple reaction tasks were unaffected by short-term chocolate supplementation in this phase. Notably, DC intake attenuated exercise-induced muscle soreness in the menstrual phase. Unlike the PLA and CON conditions, which showed significant increases in soreness 12–24 h after the workout, the DC condition showed *no* significant change in DOMS across time points and had significantly lower 12 h post-exercise soreness than the CON. This protective effect suggests that DC’s antioxidant and anti-inflammatory properties helped limit muscle damage and inflammation during menstruation [51]. Such an effect is valuable because the early-cycle hormone environment, characterized by very low sex hormones, may offer less natural anti-inflammatory protection, potentially making women more susceptible to muscle soreness. By blunting the DOMS response, DC may aid recovery during menstruation, enabling athletes to maintain training quality throughout their cycle.

Follicular phase: In the follicular phase (days after menses, with rising estrogen), DC again enhanced physical performance in the CINDY WOD. Participants achieved more repetitions with DC, outperforming both the PLA and CON conditions. This phase is often considered favorable for exercise performance due to the rise in estrogen levels, which can support muscle function and recovery [52,53]. Even so, the additional polyphenols and nutrients from DC provided further performance gains, highlighting its potential as a natural ergogenic aid [21]. No significant differences in RC or cognitive accuracy were observed with DC during the follicular phase, and HGS remained similar across conditions. These findings suggest that when women are in a hormonally neutral or improving state, characterized by minimal PMS symptoms and moderate estrogen levels, DC primarily enhances physical endurance capacity. The lack of cognitive changes could indicate that baseline cognitive function was already high in this phase, with minimal interference from hormonal symptoms, leaving less room for DC-induced improvement. It is also possible that the acute dose and timing of DC were more tuned to affecting exercise metabolism than cognitive processing when hormonal conditions were optimal. Overall, DC’s effect in the follicular phase was evident in the strenuous full-body WOD performance, consistent with its known effects on boosting exercise energy availability (e.g., via improved mitochondrial function and blood flow, as discussed later) [54].

Luteal phase: The luteal phase (post-ovulation, high progesterone) presented a somewhat different profile. DC still significantly improved CINDY WOD scores, indicating that even when the core body temperature and metabolic rate are elevated in the high-hormone phase [55], DC can enhance high-intensity exercise performance. Interestingly, the most notable effect of DC in the luteal phase was on recovery: participants reported lower DOMS in the 72 h following the workout when supplemented with DC. In our study, only the luteal-phase DOMS showed a statistically significant reduction with DC versus CON, whereas in other phases, post-exercise soreness was similar across conditions. This phase-specific attenuation of muscle soreness may relate to progesterone’s interaction with inflammatory pathways; the luteal environment might predispose athletes to more significant inflammation or fluid retention, and the antioxidants in DC could be particularly effective under these conditions [56]. From a practical standpoint, reduced soreness indicates better recovery and increased readiness for subsequent training, a crucial consideration for athletes. No improvements in HGS were observed, reinforcing that maximal strength is unaffected by acute DC intake. Cognitive performance on the Stroop test in the luteal phase remained unchanged with DC. One explanation is that while some women experience mild cognitive/mood disturbances in the mid-luteal phase, these are typically less pronounced than premenstrual symptoms [30]. Thus, the magnitude of any DC benefit on cognition might not reach significance in the mid-luteal phase. Nonetheless, the improved physical performance and recovery in this phase align with DC’s anti-fatigue and anti-inflammatory properties reported in the literature. Cocoa flavanol supplementation has been shown to blunt exercise-induced oxidative stress and potentially delay fatigue onset [51], which could be especially beneficial when luteal-phase physiology might otherwise hinder performance (e.g., through elevated catabolism or central nervous system fatigue). Our findings in the luteal phase contribute new evidence that a simple dietary intervention like DC can offset some luteal-related performance issues (such as heightened soreness) without adverse effects on strength or cognition.

Premenstrual syndrome phase: The late luteal phase, characterized by PMS symptoms in susceptible individuals, was where DC had the most multifaceted effect. Physically, DC intake resulted in the highest CINDY WOD scores among all phases, significantly outperforming both the PLA and CON groups. This suggests that DC provided a critical boost during a phase when women often feel lethargic or weak. Psychologically and cognitively, the premenstrual phase was significantly enhanced by DC: participants exhibited faster RT and a higher CAP score on the Stroop test after DC supplementation. Neither the PLA nor CON conditions matched this cognitive performance, implying a robust effect of DC on brain function under PMS conditions. These results are particularly meaningful because the PMS phase is often associated with impairments, such as poorer concentration, slower cognitive processing, and negative mood [55]. In the CON condition, participants likely experienced the typical PMS-related dips in cognitive sharpness (and perhaps motivation), but with DC, those deficits were mitigated or even reversed. The DC group’s Stroop performance in PMS was on par with or better than their performance in non-PMS phases, indicating that DC effectively leveled out the cognitive fluctuations that normally occur across the cycle. From a physiological perspective, no significant change in HGS or DOMS was noted in PMS with DC. The lack of DOMS reduction in PMS contrasts with the results of the luteal phase. This could be because by the premenstrual days the acute exercise-induced soreness was already tapering off, or because PMS-related aches are more systemic (e.g., headaches and cramps) rather than localized muscle soreness from exercise. It is essential to note that the enhanced cognitive and CINDY WOD outcomes in the DC-PMS condition likely result from both the alleviation of PMS symptoms and the direct ergogenic effects of DC. Women with more severe PMS symptoms tend to have objectively worse cognitive performance [30], so the introduction of DC—known to improve mood and induce calming neurochemical changes [57]—provided a noticeable performance uplift. In summary, the premenstrual phase saw the most significant benefit from DC, spanning both mental and physical domains, which underscores the value of this nutritional intervention during a typically challenging time in the menstrual cycle for female athletes.

### 4.3. Cognitive Performance Across Phases

The Stroop test results in our study revealed that the influence of DC on cognitive performance was highly phase-dependent. When examining the CAP and RT across menstrual phases, a clear pattern emerged: the cognitive benefits of DC were concentrated in the premenstrual period, with negligible effects in other phases. In the menstrual, follicular, and mid-luteal phases, CAP and RT did not differ significantly between the DC, PLA, or CON conditions. This suggests that in phases where women generally feel physiologically and mentally normal, acute cocoa supplementation does not substantially alter cognitive function—possibly because baseline performance is already near optimal and there is little “deficit” for the supplement to overcome. By contrast, in the PMS phase (late luteal), DC conferred a significant cognitive advantage. Participants had quicker reactions and made fewer errors on the Stroop test with DC. In contrast, in the PLA and CON conditions, their performance during the PMS phase was comparatively slower and less accurate. Statistically, this translated into significant between-condition differences only in the premenstrual phase. These findings are consistent with the notion that cognitive function in female athletes can fluctuate with cyclical symptoms: during PMS, factors such as mood swings, irritability, and fatigue can impair tasks requiring attention and executive control [30]. By ameliorating some of these PMS-related factors, the intake of DC effectively sharpened cognitive performance. It is worth noting that all participants performed the Stroop test at baseline (before exercise) in each trial; therefore, the cognitive measures reflect the acute supplementation effects rather than post-exercise fatigue. The improvement in Stroop outcomes with DC during PMS thus suggests that nutritional modulation of cognitive processes occurs under hormonally suboptimal conditions. In practical terms, an acute dose of DC before a cognitively demanding activity (e.g., strategizing workouts or making sports decisions) may be most beneficial when an athlete is in the PMS phase. The lack of cognitive enhancement in other phases also aligns with prior research on cocoa flavonoids: while many studies show that cocoa ingestion can acutely improve cognitive function, especially in tasks of executive function or mental fatigue, these effects are often more detectable when baseline cognitive performance is challenged or lowered (such as during stress, sleep deprivation, or, in this case, PMS) [58]. Our results, therefore, refine the understanding of cocoa’s nootropic effects by introducing the menstrual phase as a significant moderator. Essentially, DC served as a cognitive countermeasure specifically designed to address PMS-related cognitive sluggishness. Across all phases, it is notable that no adverse effect on cognition was observed with DC (e.g., no worsening of Stroop interference or speed–accuracy tradeoff), which supports the safe use of DC even when cognitive demands are high. In summary, the Stroop test findings highlight that phase-specific symptoms modulate the efficacy of DC on cognition: the supplement is most effective when there is a cognitive deficit to rescue (e.g., PMS) and may be redundant when cognitive function is already unimpaired (e.g., follicular, etc.). This phase-contingent benefit aligns with a personalized nutrition approach, suggesting that female athletes may derive the maximal cognitive advantage from functional foods, such as DC, at specific times in their cycle.

### 4.4. Mechanistic Insights and Possible Explanations

The ergogenic effects observed with DC can be attributed to several biochemical and physiological mechanisms, supported by recent literature. First, DC is rich in cocoa flavonols—polyphenolic compounds such as epicatechin and catechin—which have well-documented antioxidant and vasodilatory properties [21]. Acute intake of cocoa flavanols has been shown to increase cerebral blood flow and oxygenation, leading to improved cognitive performance on demanding tasks [58]. In our study, this likely underlies the enhanced Stroop test results seen in the DC-PMS condition. By improving cerebral perfusion, DC may help counteract the slight cognitive sluggishness associated with PMS, effectively “lighting up” neural circuits that might otherwise be underactive due to hormonal-induced mood and energy fluctuations. Moreover, cocoa flavonols can increase levels of neurotrophins and neurotransmitters that support cognitive function. Chronic cocoa consumption has been linked to increased brain-derived neurotrophic factor (BDNF) and improved executive function in young adults [58]. Although our trial was acute, it is plausible that even short-term DC supplementation provided a surge in neuromodulators (e.g., dopamine or serotonin) that enhanced cognitive flexibility and reaction speed during the Stroop test.

Second, the performance benefits of DC in high-intensity exercise can be partially attributed to its effect on cardiovascular and metabolic function. Epicatechin in cocoa stimulates endothelial nitric oxide production, leading to vasodilation and improved muscle blood flow [59]. This could allow more efficient oxygen and nutrient delivery to working muscles during Cindy WOD, thereby enhancing endurance and delaying fatigue. Indeed, a similar supplement rich in flavonols was found to improve peak oxygen uptake and exercise capacity in healthy individuals by enhancing vascular function [60]. Additionally, DC provides a modest amount of simple carbohydrates and fats, which could acutely contribute to energy supply during exercise. However, the magnitude of our observed performance gains (DC vs. PLA) suggests that it is the bioactive components, rather than the macronutrient content of chocolate, driving the effect, since the placebo condition presumably matched calories without those bioactive components.

Importantly, cocoa’s antioxidant and anti-inflammatory actions explain the reduced DOMS in the luteal phase. Strenuous exercise generates reactive oxygen species and provokes inflammation, contributing to muscle soreness and performance decrements in subsequent days. Cocoa flavonols can scavenge free radicals and modulate inflammatory pathways. A recent narrative review concluded that acute and short-term cocoa flavanol supplementation attenuates exercise-induced oxidative stress, with the potential to lessen fatigue and muscle damage [51]. By extension, in our study, DC likely mitigated the oxidative muscle damage incurred during the WOD, particularly under the hormonally high luteal phase, where baseline inflammation might have been elevated. This resulted in lower perceived muscle soreness on the visual analog scale in the 48–72 h following exercise with DC. Although we did not directly measure blood biomarkers, our findings resonate with those of Corr et al. (2021), who noted that while cocoa flavanols consistently reduce oxidative stress, their effects on inflammation and functional recovery, though promising, remain variable and warrant further research [51]. It is worth noting that no reduction in DOMS was observed in the PMS group in our study, potentially because PMS-related discomfort is more systemic and not solely a result of EIMD. Conversely, the luteal phase provided a more transparent window to observe cocoa’s muscle recovery benefits in isolation.

The potential ergogenic effects of DC supplementation may be attributed to its high content of flavonoids, particularly flavonols, which are known to exert several physiological benefits relevant to athletic performance and recovery. One of the primary mechanisms involves the enhancement of NO bioavailability, which contributes to improved endothelial function and vascular dilation, ultimately increasing oxygen and nutrient delivery to skeletal muscle during high-intensity efforts [61]. These effects are significant in HIFT environments, where both aerobic and anaerobic demands are elevated [13]. Moreover, DC polyphenols possess antioxidant and anti-inflammatory properties, which may attenuate exercise-induced oxidative stress and muscle damage, facilitating faster recovery and enhancing neuromuscular performance [19]. These protective effects may be particularly beneficial during hormonally sensitive phases, such as the late luteal (premenstrual) phase, when systemic inflammation and perceived fatigue levels are often elevated [62]. In addition to peripheral effects, cocoa flavonoids and other bioactive compounds may influence central fatigue mechanisms. DC contains tryptophan, a precursor of serotonin, which plays a critical role in mood regulation and central nervous system function [24,25]. During the premenstrual phase, fluctuations in serotonin levels are associated with increased emotional reactivity and reduced cognitive performance [63]. By modulating serotonin pathways, DC intake may help alleviate some of the central symptoms of fatigue and improve cognitive and neuromuscular performance during this phase [64]. These findings support the potential role of DC as a functional nutritional strategy for female athletes, particularly those engaged in metabolically demanding training modalities such as CrossFit^®^ and during hormonally sensitive phases of the menstrual cycle.

Another mechanistic factor, particularly relevant to the improvements in the PMS phase, is the interaction of DC with neurotransmitter systems that regulate mood and pain. PMS is associated with altered serotonin and GABAergic activity in the brain [30], leading to symptoms like irritability, depression, and pain sensitivity. DC contains small amounts of bioactive compounds (e.g., tryptophan, anandamide, and phenylethylamine) that can enhance the production of serotonin and endorphins [65,66]. Furthermore, cocoa is unusually rich in magnesium—a 100 g serving of DC provides a substantial fraction of the RDA—and magnesium supplementation is known to alleviate PMS symptoms, including mood swings and cramps, by modulating neuromuscular excitability [67]. In the work of Verma and Kadam (2019), daily DC significantly reduced both premenstrual and menstrual pain in young women, functioning comparably to non-steroidal anti-inflammatory drugs [24]. They attributed this analgesic effect, in part, to chocolate’s ability to increase γ-aminobutyric acid (GABA) activity and serotonin levels in the central nervous system, producing anxiolytic and comforting effects. Our results are in agreement: while we did not directly measure pain, the PMS-phase boost in cognitive and physical performance with DC likely reflects an improved psychophysiological state—characterized by less pain, a better mood, and calmer nerves. Ferina et al. (2023) offer a complementary perspective, describing DC as a non-pharmacological alternative for alleviating dysmenorrhea [25]. In their trial, a mere 35 mg/day of DC (a minimal dose) from the onset of menses appreciably lowered menstrual pain scores compared to controls, an effect they attributed to chocolate’s copper content facilitating endorphin release [25]. Endorphins are endogenous opioids that blunt pain perception and induce euphoria. Thus, even a tiny amount of DC can trigger a cascade of neurochemical changes—elevated endorphins, activated GABA_A receptors, and serotonin release—that collectively improve mood and reduce pain. Translating those findings to our context, the 30 g/day of DC given to our athletes in PMS likely acted to counteract the typical PMS symptoms (e.g., by releasing endorphins that inhibit pain impulses and by calming neural activity via GABA), thereby enabling the athletes to perform at a higher level both mentally and physically. This mechanism is a key reason why DC proved most effective in the premenstrual phase, as it addresses an underlying deficiency (low serotonin and GABA, high stress) that is specific to PMS. In phases where such imbalances are absent (e.g., mid-cycle), those exact mechanisms would not have as dramatic an observable effect.

Lastly, we consider why HGS was unaffected by DC in all phases. HGS is a measure of maximal voluntary force. It is primarily limited by neuromuscular activation and muscle fiber recruitment, which are less likely to be influenced by short-term nutritional interventions. Unlike endurance or cognitive tasks, which can benefit from improved blood flow or slight changes in neurotransmission, a maximal-strength effort relies on the activation of peak motor units [68]. Our findings concur with the absence of literature showing acute chocolate effects on strength. The improvement seen in dynamic exercise performance (CINDY WOD) but not static strength suggests that DC’s ergogenic components primarily enhance submaximal, sustained efforts—likely by delaying fatigue and improving efficiency—rather than increasing maximal force output. This pattern is mirrored in other research on polyphenol supplementation; for instance, beetroot juice (rich in nitrates) often improves endurance more than one-repetition maximum (1RM) strength due to similar vasodilatory and fatigue-delaying pathways [69]. DC’s lack of effect on HGS also reinforces that the participants were not experiencing any stimulant-like benefit (as caffeine would confer). It rules out the possibility that simple sugar intake was a factor, since a sugar rush could theoretically enhance short-burst strength, but we did not observe that. Therefore, the mechanistic takeaway is that DC’s bioactive nutrients target metabolic and neural aspects of performance that translate to improved endurance and cognitive function while leaving maximal strength untouched. In summary, multiple mechanisms likely acted in concert: (1) cocoa flavanols improved vascular and metabolic responses (boosting WOD performance and recovery), (2) antioxidants reduced oxidative stress (diminishing DOMS in the luteal phase), and (3) neuroactive compounds in chocolate modulated neurotransmitters (alleviating PMS symptoms and enhancing cognitive speed/accuracy). Contemporary studies support these insights and provide a biological rationale for the phase-specific benefits we observed. Such mechanisms highlight DC as a unique nutraceutical that can simultaneously target peripheral physiological function and central nervous system state—an appealing combination for female athletes navigating the complexities of the menstrual cycle’s effects on their training.

While this study’s findings are promising, it is essential to note that they were derived from a relatively small and homogeneous sample of trained female athletes. Future studies should consider larger sample sizes and multicentric designs involving more diverse athletic populations to enhance the external validity and explore broader applicability.

### 4.5. Limitations and Future Research Directions

This study offers valuable insights into the short-term effects of DC supplementation on exercise recovery and cognitive performance across various phases of the menstrual cycle in trained female CrossFit^®^ athletes. However, several limitations should be acknowledged. First, although this study met the required sample size estimated through a priori power analysis (α = 0.05, power = 0.80, effect size = 0.5), the relatively small cohort (*n* = 15) may still limit the generalizability of our findings. The crossover design helped control for individual variability and enhanced internal validity, reducing the likelihood of Type I errors. However, the homogeneity of the sample—trained eumenorrheic female CrossFit^®^ athletes—restricts extrapolation to broader athletic or clinical populations. Future research should involve larger, more diverse samples through multicentric trials to confirm these findings and explore inter-individual variability in response to dark chocolate supplementation across menstrual phases. Furthermore, this study exclusively included eumenorrheic women with regular menstrual cycles; therefore, the findings cannot be generalized to women with irregular or anovulatory cycles, who may exhibit different hormonal profiles and physiological responses. Second, menstrual phase classification was based on self-reporting via the PSST tool and basal body temperature tracking, which, while reliable, lacks the accuracy of hormonal assays (e.g., estradiol and progesterone concentrations) that could have verified phase timing with greater precision. Third, the intervention focused on acute (3-day) supplementation within each menstrual phase, which may not capture the cumulative or long-term effects of chronic DC intake. Whether sustained consumption yields enhanced benefits or leads to adaptation or tolerance remains unclear. Additionally, although the placebo was matched in appearance and macronutrient content, the unique sensory properties of DC may have compromised blinding, and this was not formally evaluated through post-intervention assessments. From a methodological standpoint, while the selected functional (CINDY WOD and HGS) and cognitive (Stroop test) measures are relevant to CrossFit^®^, they represent only a subset of performance domains. Variables such as explosive power, reaction decision making under fatigue, or hormonal biomarkers were not assessed.

Future research should expand upon these findings by including larger and more diverse female populations, including untrained individuals and athletes from other sports. The integration of biochemical markers (e.g., IL-6, TNF-α, and cortisol) and hormonal profiles would enhance mechanistic understanding. Exploring different doses, durations, or timing of supplementation—especially during hormonally sensitive phases like PMS—would help define optimal protocols. Finally, future studies should consider assessing subjective factors such as mood, sleep quality, and perceived exertion, which are known to fluctuate with hormonal changes and could interact with supplementation effects.

## 5. Conclusions

This study provides preliminary evidence that short-term supplementation with DC may positively influence athletic performance, cognitive function, and muscle recovery in female CrossFit^®^ athletes, particularly during hormonally sensitive phases such as the luteal and premenstrual stages. The improvements in functional performance (CINDY WOD) and reduction in DOMS suggest a beneficial role of DC in modulating inflammatory and neuromuscular responses. Additionally, enhancements in reaction time during the PMS phase highlight its potential impact on cognitive resilience under hormonal stress. These findings support the integration of phase-specific nutritional strategies to optimize performance and recovery in female athletes. Practically, consuming 30 g/day of 85% DC for three days during these phases may provide a simple and accessible strategy to support recovery and training quality. However, further research involving larger samples, hormonal profiling, and long-term interventions is needed to confirm and extend these observations.

## Figures and Tables

**Figure 1 nutrients-17-01374-f001:**
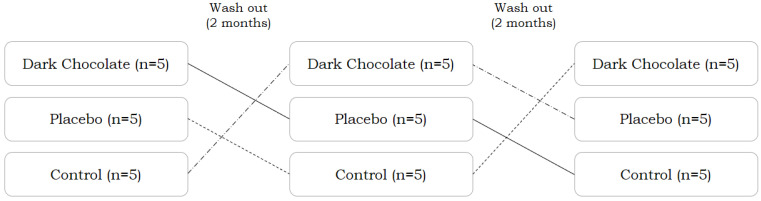
Crossover and double-blinded study design in three conditions.

**Figure 2 nutrients-17-01374-f002:**
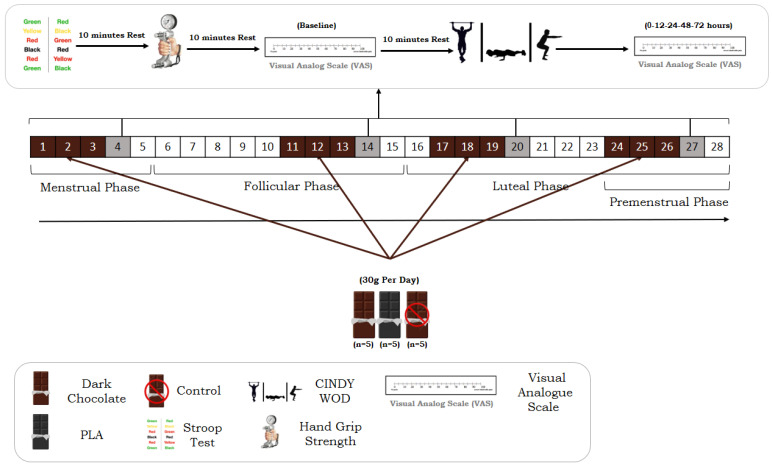
The protocol of taking supplements and performing tests. VAS: visual analog scale, CINDY WOD.

**Figure 3 nutrients-17-01374-f003:**
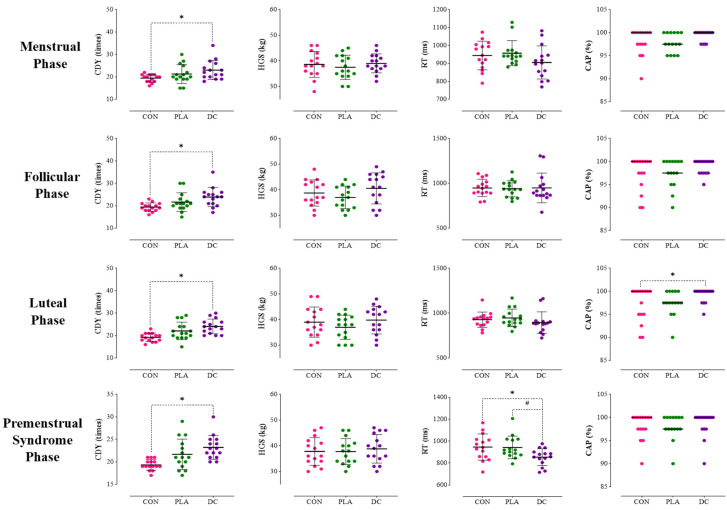
Individual responses, means, and standard deviations of the CDY, HGS, RT, and CAP, in the three conditions and four different menstrual phases. CON: control, PLA: placebo, DC: dark chocolate, CDY: CINDY WOD, HGS: handgrip strength, RT: reaction time, CAP: correct answer percentage. *: Significant difference compared to the CON (*p* < 0.05). ^#^: Significant difference compared to the PLA (*p* < 0.05).

**Figure 4 nutrients-17-01374-f004:**
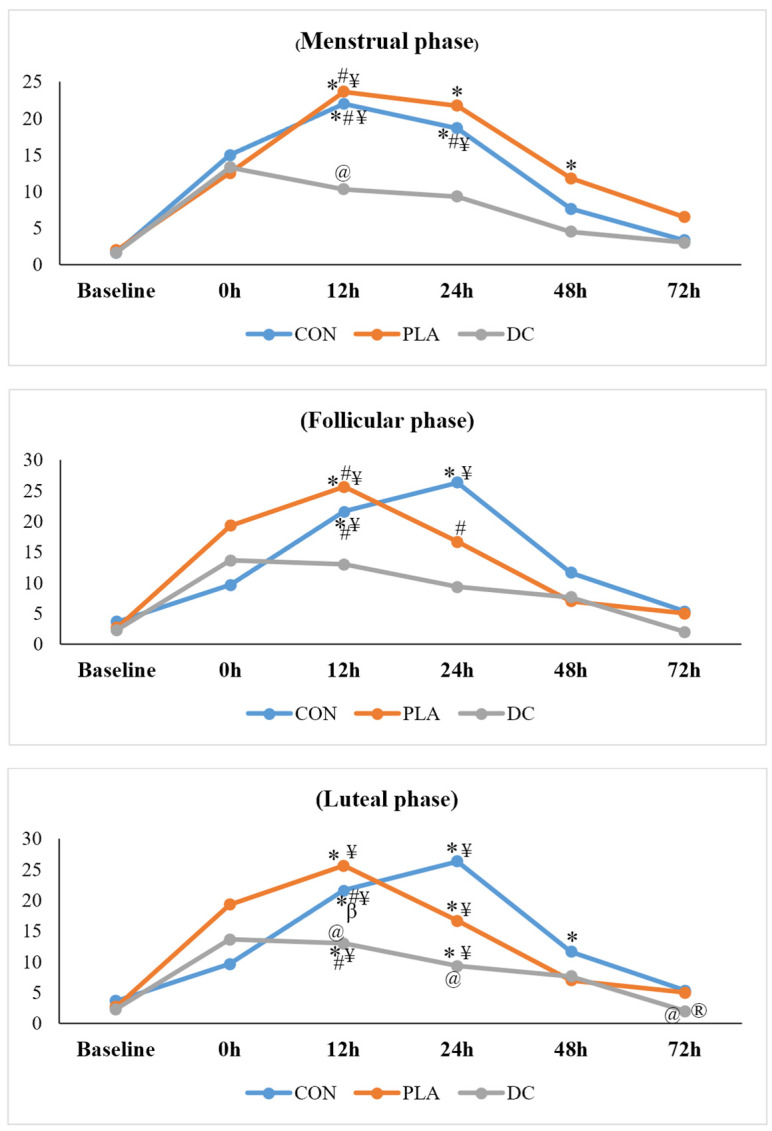
Changes in average DOMS across different menstrual phases and under various supplementation conditions. CON: control; PLA: placebo; DC: dark chocolate; baseline: before CINDY WOD; 0 h: immediately after CINDY WOD; 12 h: 12 h after CINDY WOD; 24 h: 24 h after CINDY WOD; 48 h: 48 h after CINDY WOD; 72 h: 72 h after CINDY WOD. *: Significant difference compared to the baseline (*p* < 0.05). ^β^: Significant difference compared to 0 h (*p* < 0.05). ^#^: Significant difference compared to 48 h (*p* < 0.05). ^¥^: Significant difference compared to 72 h (*p* < 0.05). ^@^: Significant difference compared to the CON at the same time (*p* < 0.05). ^®^: Significant difference compared to the PLA at the same time (*p* < 0.05).

**Table 1 nutrients-17-01374-t001:** The anthropometric data of participants.

Characteristic	Mean ± SD (*n* = 15)
**Age (years)**	22.9 ± 0.5
**Height (cm)**	159.6 ± 1.1
**Weight (kg)**	56.9 ± 1.7
**BMI (kg/m^2^)**	22.2 ± 0.5

**Table 2 nutrients-17-01374-t002:** Mean and standard deviation (mean ± SD) of measured variables (*n* = 15).

Phase	Variables	CON	PLA	DC
**Menstrual Phase**	**CDY (Times)**	19.46 ± 1.72	21.26 ± 4.18	23.00 ± 4.27
	**HGS (kg)**	38.60 ± 5.03	37.46 ± 4.71	39.00 ± 3.70
	**RT (ms)**	944.06 ± 79.62	957.00 ± 69.92	905.33 ± 92.14
	**CAP (%)**	98.00 ± 2.86	97.83 ± 2.08	99.33 ± 1.14
**Follicular Phase**	**CDY (Times)**	19.40 ± 1.88	21.60 ± 4.13	23.86 ± 4.27
	**HGS (kg)**	38.73 ± 5.10	37.00 ± 4.39	40.53 ± 6.08
	**RT (ms)**	948.13 ± 95.94	939.53 ± 90.29	948.86 ± 165.62
	**CAP (%)**	97.33 ± 3.71	97.50 ± 3.13	98.83 ± 1.59
**Luteal Phase**	**CDY (Times)**	19.20 ± 1.89	22.06 ± 3.91	24.06 ± 3.26
	**HGS (kg)**	39.00 ± 5.90	37.00 ± 4.72	39.80 ± 5.38
	**RT (ms)**	926.93 ± 85.30	946.06 ± 97.91	894.86 ± 119.78
	**CAP (%)**	91.00 ± 3.80	97.50 ± 2.67	99.16 ± 1.54
**Premenstrual Syndrome Phase**	**CDY (Times)**	19.33 ± 1.17	21.66 ± 3.39	23.20 ± 2.62
	**HGS (kg)**	37.80 ± 5.40	37.73 ± 5.04	38.80 ± 5.54
	**RT (ms)**	946.73 ± 119.49	943.06 ± 103.00	854.33 ± 76.86
	**CAP (%)**	98.00 ± 2.86	98.00 ± 2.70	98.50 ± 2.80

CON: control, PLA: placebo, DC: dark chocolate, CDY: Cindy test, HGS: handgrip strength, RT: reaction time, CAP: correct answer percentage, kg: kilogram, ms: millisecond.

**Table 3 nutrients-17-01374-t003:** Mean and standard deviation (mean ± SD) of the DOMS in each condition and phase (*n* = 15).

Phase	Variables	CON	PLA	%RC_PLA/CON_	DC	%RC_DC/CON_	%RC_DC/PLA_
**Menstrual Phase**	**DOMS Baseline (mm)**	1.66 ± 4.49	2.00 ± 3.68	20.48%	1.66 ± 3.61	0.00%	−17.00%
	**DOMS 0 h (mm)**	15.00 ± 17.32	12.53 ± 16.20	−16.47%	13.33 ± 14.71	−11.13%	6.38%
	**DOMS 12 h (mm)**	22.00 ± 19.06	23.66 ± 16.19	7.55%	10.33 ± 13.15	−53.05%	−56.34%
	**DOMS 24 h (mm)**	18.66 ± 15.40	21.73 ± 20.26	16.45%	9.33 ± 15.33	−50.00%	−57.06%
	**DOMS 48 h (mm)**	7.66 ± 9.42	11.80 ± 11.68	54.05%	4.53 ± 10.60	−40.86%	−61.61%
	**DOMS 72 h (mm)**	3.33 ± 6.17	6.53 ± 9.61	96.10%	3.00 ± 5.91	−9.91%	−54.06%
**Follicular Phase**	**DOMS Baseline (mm)**	3.66 ± 8.12	2.66 ± 4.16	−27.32%	2.33 ± 3.71	−36.34%	−12.41%
	**DOMS 0 h (mm)**	9.66 ± 11.41	19.33 ± 24.48	100.10%	13.66 ± 23.10	41.41%	−29.33%
	**DOMS 12 h (mm)**	21.66 ± 16.54	25.66 ± 22.18	18.47%	13.00 ± 16.77	−39.98%	−49.34%
	**DOMS 24 h (mm)**	26.33 ± 16.95	16.66 ± 15.54	−36.73%	9.33 ± 16.78	−64.57%	−44.00%
	**DOMS 48 h (mm)**	11.66 ± 12.34	7.00 ± 9.96	−39.97%	7.66 ± 15.79	−34.31%	9.43%
	**DOMS 72 h (mm)**	5.33 ± 8.12	3.33 ± 8.38	−37.52%	2.00 ± 3.68	−62.48%	−39.94%
**Luteal Phase**	**DOMS Baseline (mm)**	1.00 ± 2.80	12.66 ± 13.34	1166.00%	1.66 ± 5.23	66.00%	−86.89%
	**DOMS 0 h (mm)**	12.33 ± 13.87	22.33 ± 16.88	81.10%	13.66 ± 17.77	10.79%	−38.83%
	**DOMS 12 h (mm)**	29.00 ± 15.49	23.00 ± 17.60	−20.69%	16.33 ± 14.32	−43.69%	−29.00%
	**DOMS 24 h (mm)**	23.66 ± 13.94	14.33 ± 13.47	−39.43%	10.66 ± 9.03	−54.95%	−25.61%
	**DOMS 48 h (mm)**	14.66 ± 12.60	7.66 ± 9.42	−47.75%	5.66 ± 8.83	−61.39%	−26.11%
	**DOMS 72 h (mm)**	10.33 ± 11.72	1.66 ± 5.23	−83.93%	0.66 ± 1.75	−93.61%	−60.24%
**Premenstrual Syndrome Phase**	**DOMS Baseline (mm)**	1.00 ± 2.80	1.66 ± 3.61	66.00%	2.00 ± 3.68	100.00%	20.48%
	**DOMS 0 h (mm)**	14.66 ± 17.67	14.33 ± 14.37	−2.25%	15.66 ± 13.99	6.82%	9.28%
	**DOMS 12 h (mm)**	19.33 ± 16.78	16.00 ± 16.49	−17.23%	15.00 ± 12.81	−22.40%	−6.25%
	**DOMS 24 h (mm)**	14.66 ± 13.15	17.66 ± 18.88	20.46%	11.66 ± 12.77	−20.46%	−33.98%
	**DOMS 48 h (mm)**	7.33 ± 8.63	9.66 ± 12.31	31.79%	3.33 ± 5.87	−54.57%	−65.53%
	**DOMS 72 h (mm)**	4.33 ± 5.93	4.00 ± 7.12	−7.62%	0.66 ± 2.58	−84.76%	−83.50%

CON: control; PLA: placebo; DC: dark chocolate; RC: relative change; baseline: before CINDY WOD; 0 h: immediately after CINDY WOD; 12 h: 12 h after CINDY WOD; 24 h: 24 h after CINDY WOD; 48 h: 48 h after CINDY WOD; 72 h: 72 h after CINDY WOD.

**Table 4 nutrients-17-01374-t004:** Pairwise comparisons in the three supplementation conditions (*n* = 15).

Phases			PLA		DC	
			CON	DC	CON	PLA
**Menstrual Phase**	**CDY** **(Times)**	**MD**	1.80	−1.73	3.53	1.73
		**Sig**	0.433	0.513	0.017	0.513
		**95%CI**	−1.36–4.96	−4.99–1.53	0.581–6.48	−1.53–4.99
	**HGS** **(kg)**	**MD**	−1.13	−1.53	0.400	1.53
		**Sig**	0.943	0.634	1.000	0.634
		**95%CI**	−4.08–1.81	−4.71–1.64	−3.53–4.33	−1.64–4.71
	**RT** **(ms)**	**MD**	12.93	51.66	−38.73	−51.66
		**Sig**	1.000	0.175	0.740	0.175
		**95%CI**	−49.41–75.27	−16.49–119.82	−125.80–48.33	−119.82–16.49
	**CAP** **(%)**	**MD**	−0.16	−1.50	1.33	1.50
		**Sig**	1.000	0.042	0.359	0.042
		**95%CI**	−2.41–2.07	−2.95–−0.04	−0.85–3.51	0.04–2.95
**Follicular Phase**	**CDY** **(Times)**	**MD**	−1.73	−3.53	1.80	3.53
		**Sig**	0.307	0.077	0.008	0.077
		**95%CI**	−1.21–5.61	−4.73–0.200	−1.14–7.78	−0.200–4.73
	**HGS** **(kg)**	**MD**	12.49	−4.14	16.64	4.14
		**Sig**	0.91	0.114	0.797	0.114
		**95%CI**	−3.69–0.224	−7.72–0.659	−2.42–6.02	−0.65–7.72
	**RT** **(ms)**	**MD**	−8.60	−9.33	0.73	9.33
		**Sig**	1.000	1.000	1.000	1.000
		**95%CI**	−56.57–39.37	−118.23–99.57	−111.64–113.11	−99.571–118.238
	**CAP** **(%)**	**MD**	0.167	−1.33	1.50	1.33
		**Sig**	1.000	0.493	0.323	0.493
		**95%CI**	−3.04–3.37	−3.80–1.13	−0.872–3.87	−1.13–3.80
**Luteal Phase**	**CDY** **(Times)**	**MD**	2.86	−2.00	4.86	2.00
		**Sig**	0.096	0.213	0.000	0.213
		**95%CI**	−0.40–6.13	−4.78–0.782	2.32–7.40	−0.782–4.782
	**HGS** **(kg)**	**MD**	−2.00	−2.80	0.800	2.80
		**Sig**	0.605	0.207	1.000	0.207
		**95%CI**	−6.05–2.05	−6.66–1.06	−3.58–5.18	−1.06–6.66
	**RT** **(ms)**	**MD**	19.13	51.20	−32.06	−51.20
		**Sig**	1.000	0.611	1.000	0.611
		**95%CI**	−40.36–78.63	−53.13–155.53	−131.47–67.34	−155.53–53.13
	**CAP** **(%)**	**MD**	0.50	−1.66	2.16	1.66
		**Sig**	1.000	0.108	0.080	0.108
		**95%CI**	−1.90–2.90	−3.61–0.285	−0.21–4.54	−0.28–3.61
**Premenstrual Syndrome Phase**	**CDY** **(Times)**	**MD**	2.33	−1.53	3.86	1.53
		**Sig**	0.087	0.157	0.002	0.157
		**95%CI**	−0.274–4.94	−3.49–0.431	1.48–6.25	−0.431–3.49
	**HGS** **(kg)**	**MD**	−0.06	−1.06	1.00	1.06
		**Sig**	1.000	1.000	1.000	1.000
		**95%CI**	−2.77–2.64	−4.32–2.18	−3.24–5.24	−2.18–4.32
	**RT** **(ms)**	**MD**	−3.66	88.73	−92.40	−88.73
		**Sig**	1.000	0.002	0.010	0.002
		**95%CI**	−71.17–63.83	32.14–145.32	−163.12–−21.67	−145.32–−32.14
	**CAP** **(%)**	**MD**	0.00	−0.50	0.50	0.50
		**Sig**	1.000	1.000	1.000	1.000
		**95%CI**	−2.39–2.39	−2.99–1.99	−2.32–3.32	−1.99–2.99

CON: control, PLA: placebo, DC: dark chocolate, CDY: Cindy test, HGS: handgrip strength, RT: reaction time, CAP: correct answer percentage, kg: kilogram, ms: millisecond, MD: mean difference, CI: confidence interval.

## Data Availability

The original contributions presented in this study are included in the article. Further inquiries can be directed to the corresponding author.

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
