# Peer review of "Dark Chocolate Mitigates Premenstrual Performance Impairments and Muscle Soreness in Female CrossFit® Athletes: Evidence from a Menstrual-Phase-Specific Trial"

_nutrients, 2025, doi:10.3390/nu17081374_

Round 1

Reviewer 1 Report

Comments and Suggestions for Authors

This manuscript addresses a highly pertinent and under-explored area of sports nutrition, investigating the effects of dark chocolate (DC) supplementation on exercise performance, cognitive function, and muscle soreness in trained female CrossFit® athletes across different menstrual cycle phases. The study design—a randomized, double-blind, placebo-controlled, cross-over trial—is rigorous and methodologically sound. The authors are to be commended for targeting an athletic female population, which remains relatively underrepresented in sports science research.

The work is relevant and offers valuable insights into phase-specific nutritional strategies. However, there are areas that warrant revision to strengthen the scientific narrative, increase clarity, and better contextualize the findings.

Major Comments

  1. Introduction 

    The introduction provides a solid overview of the hormonal fluctuations and their impact on female athletes. Nevertheless, it would benefit from further expansion regarding the physiological adaptations induced by high-intensity functional training (HIFT), such as CrossFit®, and how nutritional strategies might support these adaptations.

    In this regard, I strongly recommend the authors consider incorporating the following recent reference to broaden the context and enhance the relevance of their discussion:

    Moscatelli F., et al. Aerobic and Anaerobic Effect of CrossFit Training: A Narrative Review. Sport Mont. 2023, 21(1), pp. 123–128.

    This narrative review offers a comprehensive examination of both aerobic and anaerobic adaptations to CrossFit® training, with specific attention to metabolic and neuromuscular responses. Integrating this perspective would help bridge the gap between the physiological demands of the training modality used in your study and the potential role of targeted nutritional interventions, such as dark chocolate supplementation.

  2. Discussion – Mechanistic Depth

    The manuscript would benefit from a deeper exploration of the mechanistic pathways by which dark chocolate flavonoids and associated bioactive compounds influence neuromuscular performance and recovery. While the manuscript touches on polyphenol-mediated effects, I encourage the authors to expand on the role of nitric oxide bioavailability, endothelial function, and potential neuromodulatory effects, especially during hormonally sensitive phases.

    Additionally, consider discussing central fatigue mechanisms, particularly the impact of serotonin precursors found in cocoa, which may play a role in cognitive performance improvements observed in the PMS phase.

  3. Statistical Power and Generalizability

    Although the study is well-conducted, the small sample size (n=15) inherently limits generalizability. While this is acknowledged, I suggest the authors explicitly discuss this limitation in the Discussion section, noting that larger, multicentric trials are necessary to validate these findings and explore inter-individual variability.

  4. Presentation of Results

    Figures and tables are generally clear; however, I recommend the following improvements:

    • Ensure all figures have uniform formatting, particularly regarding significance indicators (asterisks, symbols, etc.).

    • In Table 3, consider providing absolute and relative changes alongside means and standard deviations for DOMS, to better convey the magnitude of effects.

  5. Abstract

    The abstract is comprehensive but overly detailed for the format. Consider streamlining the description of the results while preserving key findings, especially focusing on the most impactful results (e.g., improvement in reaction time during PMS and reduction of DOMS in the luteal phase).

Minor Comments

  • Ensure consistency in terminology: "dark chocolate" should be abbreviated consistently as "DC" throughout the manuscript.

  • Abbreviation list: Provide a comprehensive list of abbreviations at the beginning or end of the manuscript for reader clarity.

  • The reference section requires minor formatting corrections to align fully with the journal's style guidelines.

  • Ethical considerations: Please include the clinical trial registration number if applicable.

  • Proofreading: The manuscript would benefit from a careful language review to eliminate minor typographical errors and improve fluency.

Reviewer 2 Report

Comments and Suggestions for Authors

This is a well written, well design study

Just a few comments

Pg 2: should Premenstrual Syndrome be lowercased.  From a "Google" search: In general, do not capitalize the names of diseases, disorders, therapies, treatments, theories, concepts, hypotheses, principles, models, and statistical procedures.

Glad you did a GPower estimation. You collected enough subjects

Good use of figures/tables

Limitations

  1. You write that your sample size was small.  But didn't it meet your sample size estimation?  So didn't it have enough power then?  If it didn't have enough power wouldn't you have a lot of Type I errors?
  2. Would a limitation be that you couldn't generalize findings to women who had irregular cycles?
